# Changing Lanes: Towards Symbolizing Visualizations

Vidya Setlur and Bridget Cogley

Fig. 1. Icons and symbols in modern pop culture. Row-wise from left to right: dining, slippery road, resist, wash hands, wear a mask, C for Croissant in an alphabet card, all-gender bathroom, shh - be quiet!, "I love you" in ASL, finger hearts, pride flag, party emoji, and the COVID-19 curve.

**Abstract**— The data visualization community often focuses on identifying and putting best practices into action when thinking about chart and dashboard authoring. These practices often incorporate recommendations informed by human perception and are often prescriptive around precision at the expense of semantics. Drawing from the design of meaningful icons, symbols, and imagery, we argue that data visualization needs to be made more accessible to the general population. By focusing more on intention rather than just precision, can we symbolize visualizations and change the lanes for both the researcher and practitioner community?

**Index Terms**—Symbols, icons, register, visual design, semantics, intent.

◆

## 1 INTRODUCTION

As humans, we seek meaning, connection, certainty, and clarification in the world around us - whether it is looking at signs as we drive, finding a meal to order by scanning a menu marked with icons, or trying to understand the pointed gestures of our friend showing directions. Sports uniforms, company logos, and traffic signs (as seen in Figure 1) are all examples of symbols. They serve as pictographic representations of intent or purpose.

*Semantics* is a concept that formalizes and describes the meaning and understanding of the various relationships and patterns in the world [2]. The effective depiction of a symbol or an icon often depends on how semantically resonant the image is to the information it represents.

- *Vidya Setlur is with Tableau Research. E-mail: vsetlur@tableau.com.*
- *Bridget Cogley is with Versalytix. E-mail: bcogley@versalytix.com.*

The use of icons in charts depends on various factors, including task, how representative they are of the underlying data, and their general recognizability [7].

Charts abstract information, making it easier for people to see patterns at a distance, compare, and extrapolate. Icon encodings are graphical elements that are often used to visually represent the semantic meaning of marks for categorical data. Assigning meaningful icons to display elements helps the user perceive and interpret the visualization easier. These encodings can be effective in enabling visual analysis because they are often rapidly and efficiently processed by the preattentive visual system [8]. The human visual system spatially categorizes these icons in order to create a meaningful understanding of the visualization.

However, these mechanisms are far from perfect. While we can detect patterns, we do not always see or interpret the same things. Some of this goes back to our mind, which will fill in patterns to better align with what we expect. Charts that ask you to draw your assumptions

are powerful tools to combat some of these cognitive jumps. They force us to see what we expect compared to what the data shows. By considering semantics as part of the visualization authoring process, we can take a visual abstraction of data, agree on its meaning, and put it to use as a shared symbol.

## 2 "NOT MY DESSERT"

Emerging practice professions rely on hard demarcations between those immersed in the field of visualization and those outside it. Pie and donut charts, much aligned and dispensed with vigor, serve as a clear, bright line between those who design well and those who do not. There has been a lot of discussion about the validity of these charts as a meaningful way to convey information that could be interpreted. This is partly due to Cleveland and McGill's findings of the angle being evaluated at a pretty low accuracy level [3]. Skau and Kosara [9] decided to focus their studies on pie and donut charts and perhaps show that we perhaps should not be so quick to dish on our dessert.

How we communicate invites people to enter the conversation or wholly avoid it. *Register* [5] defines both the formality of interaction as well as the familiarity of the participants. This paper is a register-exercise, designed to provoke but play well in the academic circuit. As such, there are expectations as to how ideas are expressed and are allowed to provoke. Visualizations use register. Pie charts can communicate part-to-whole relationships very effectively and serve as a way of providing a bit of refreshment to start the meal.

As Cleveland and McGill noted, "The ordering of the tasks does not result in a precise prescription for displaying data, but rather is a framework within which to work." Start with the visual fundamentals, but there is much more to what makes a visualization effective, and we argue that it may be ok to "break" some rules. And yes, sometimes, pie charts may be the right choice.

## 3 FLATTENING THE CURVE

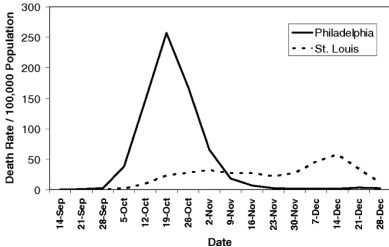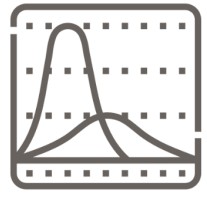

Fig. 2. COVID-19 curve. Left: The original chart. Right: The derivative showing an iconic visual.

In 2007, researchers studying the non-pharmaceutical interventions used in the 1918 flu created a chart showcasing the scenarios between two cities [4]. The original chart used a solid line to represent Philadelphia and a dotted line for St. Louis, the city with interventions. The graph is telling - not only did Philadelphia spike harder and faster - it had more cases overall. St. Louis, a city that instituted masks and limited events, experienced a gentler slope, with maximum caseloads reaching one-fifth of what Philadelphia experienced.

The chart created the visual language for the "Flatten the Curve" graphic used today (Figure 2). As an icon, it has been abstracted into two rounded area charts. The charts speak not to the literal numbers but the general trend. It adapts the representation from the original line charts used to an area chart in order to make it useful to the masses.

As news organizations visualized COVID-19 data, they had to overcome hurdles associated with displaying ambiguous data. Case report dates varied, with weekend data delayed until the following Monday for reporting. This affected the visual pattern. Defining a case varied across geopolitical lines, deaths were not always directly listed as pandemic-related and the delayed relationships between onset and death needed clarification for public understanding.

Beyond the literal example, this symbol extracted a proven pattern beyond a literal example to that of a rally cry. We could use it to highlight healthcare system limits and the goal of delaying cases - we knew they would come - to perhaps when we had more capacity and more pharmaceutical options. Its ambiguity allowed its broad use but relied on clarification of what it showed.

The COVID-19 pandemic led to a proliferation of visualizations: analysts showed daily counts, rolling averages, accumulated totals over time, and a myriad of ratios in a variety of ways. Yet, we did not take away the same understanding of the data, and a small network leveraged this ambiguity to create more confusion. As we abstract forms, we rely on shared semantic understandings of what is encoded in the chart.

## 4 MAPS AND WAYFINDING

A map is another unique form of symbolic representation but is specialized for space. It depicts spatial relationships with objects in the geographic and physical landscape. By scaling down to a natural size conducive for reading, a map allows the viewer to see far more information in one glance than could ever be possible from ground level without the need for any pictorial realism.

Maps have inspired the early roots of information visualization by depicting information as a communication process [6]. While details of these depictions vary, all maps share a basic structure with an information source identified by the cartographer who determines what and how to represent that information to the reader. The map serves as one of many potential representations of information in space that allows the reader to access its meaning. The process of communicating spatial information has evolved as a system with information inputs, transmission, and reception of that information to the intended audience.

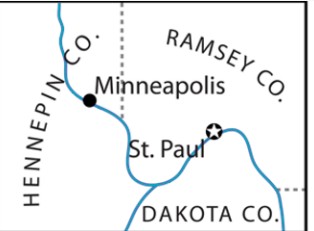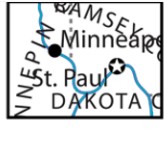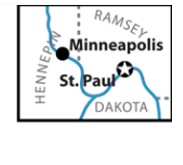

Fig. 3. A generalization example: Large cities are made more prominent (e.g., larger labels in bold) than smaller cities; interstate highways are made more prominent (e.g., lines are thicker) than local streets.

Cartographers have found ways to make maps more legible and useful. They often simplify or eliminate less semantically important features, exaggerate more important ones, and resolve visual clutter to improve information quality. This is a process known as *generalization* (as shown in Figure 3). Route maps have succinctly applied various forms of generalization to effectively provide navigational direction. Generalization helps emphasize what is important and deemphasize the unimportant, showing points of interest, turns, and interactions such as pan and zoom. Mapmakers make explicit decisions about which aspects of the route are most relevant for a navigator to understand and follow the route. They use a variety of cartographic generalization techniques to help improve the clarity of the map and emphasize only the most important information [1].

## 5 NEED FOR MORE ROAD SIGNS

Road signs are effective only when people clearly understand their meaning. A simple, yet effective universal language system has evolved over time, symbolizing key information communicated on the road. The design of red for example, is iconic for prohibited actions and items when used as a circle with a slash and stop when used as an octagon. Two lines meeting are also strongly associated with the concept of lanes merging. A cross and tracks depict iconicity for trains, while a deer on a yellow diamond is iconic for a warning of deer crossing the road. Directionality is represented by arrows that are read from bottom

to top (Figure 4) with a number nestled within the space of the curve to indicate why one should slow down. The register of these signs is appropriate for pacing traffic.

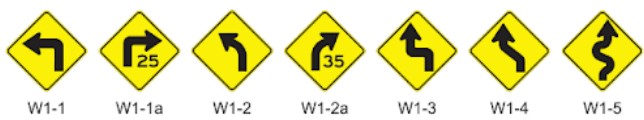

Fig. 4. A set of road sign symbols showing turn directions and speeds.

By careful consideration of different shapes, colors, and words, the register for these road signs needs to fulfill a need, command attention, convey a clear and simple meaning, command the respect of the road, and give adequate time for proper response [10]. Misinterpretations and partial interpretations of road signs indicate that the linguistic features are not iconic.

Similarly, register affects the pacing and presentation of a visualization. One really complex chart may suffice on its own but may need to be broken out into separate pieces when lowering the register. Visualizations from New York Times You-Draw-It series often teach those interacting with the chart how to use it first. They may include supports such as key dots to draw a line through to include. Other visualizations may include a highly annotated overlay to help audiences understand the chart. Some provide several registers, allowing the information to be seen with a complex chart, or switched to a layout with a lower register and several charts. In short, they have a conversation at a lower register before presenting content at a higher register. These methods prepare the audience by engaging them and giving them the tools to understand charts that may feel like jargon otherwise.

Within the analysis, chart choices can be informed by register, taking into account the formality and familiarity of the audience. As we examine displaying amounts over time in Figure 5, we may find the area chart draws undue emphasis, while the line chart is less comfortable to the audience. Blending the two lowers the register by preserving the anchor to zero while still drawing the primary emphasis to the line.

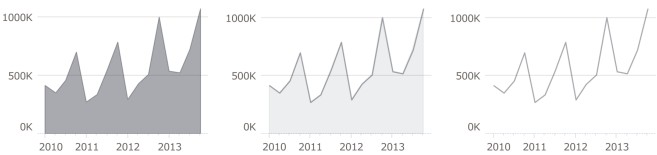

Fig. 5. Adjusting the register of a chart

By lowering the register, we ease access to the information by providing additional guardrails. Lines may be hard for some readers to navigate. The light shading can help introduce what the line represents while still providing a solid anchor.

An analogous concept is the diachronic process of language where words evolve over time, often reducing or smoothening concepts for faster efficiency. Per Skyer, "Words, concepts, and pragmatics themselves evolve and shift given new mediums of expression." "Flatten the curve" as a chart-bound graphic preserves certain aspects, regardless of the media. Similarly, map generalization focuses on what is important for wayfinding by deemphasizing and simplifying what is less important.

Let us consider another example where symboling visualizations can be effective. Figure 6 shows how a speed-limit road sign can influence the design and formatting of Big Ass Numbers (BANs). A common issue with BANs is that they are not formatted well enough to be effective, or even dynamic. By considering the aesthetics use of space to display the number prominently, a reader would only need to look up the number as they become familiar with the system.

We can then identify ways to extend the semantics and the author's intentionality in the design process. A dashboard containing this numerical symbol (Figure 7) shows the issues around customer traffic to two

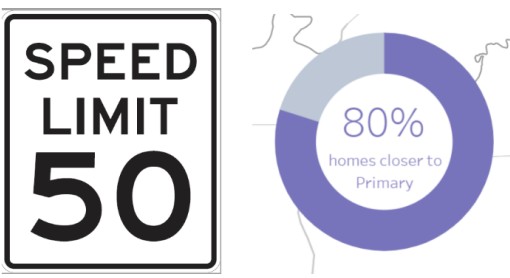

Fig. 6. Signposting for data. Left: A speed limit sign. Right: A BAN symbol.

cafes - Primary and Secondary. A two-tone color is used consistently across the Primary and Secondary items. The key takeaway is clear from the title with deliberate use of space. The content visually aligns with the takeaway problem statement, and the charts fill the space in a manner to communicate the spatial context to the reader. The content in the dashboard is laid out such that the totals for each cafe exist on one side rather than cluttering the entire space.

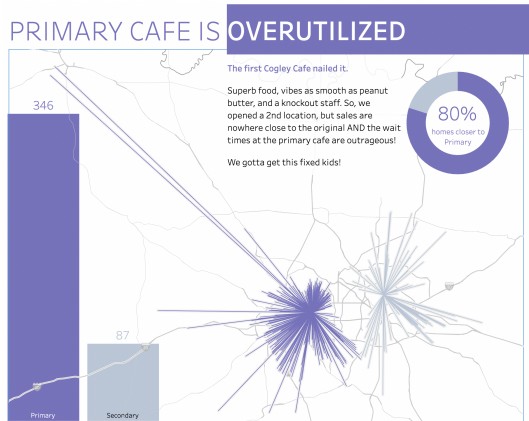

Fig. 7. Dashboard example designed to symbolize the sales at two cafes

We hope that through this provocation, the visualization community can be inspired to consider semantics and intent as first-class citizens in the visual design process. The use (and misuse) of these design choices affect how information is communicated and disseminated in our society. We close with some actionable takeaways for symbolizing visualizations so that they become more mainstream and accessible:

- Systemize your charts for a consistent reading experience.
- Use colors semantically
- Optimize for various viewing conditions (different devices, screens, visual and cognitive needs)
- Simplify and reduce for constrained viewing spaces
- Stage information (warning, more details) in a variety of ways and spaces
- Adjust the register to most effectively communicate in a given space
- Leverage deictic references by placing relevant information where they are easily understood (ex: speed limit within the curve) - note how "MPH" can be inferred, in part because the information has been used elsewhere)

To summarize, do not overthink design; often, simple charts when used appropriately, can be powerful. Let us use visualizations to take people on analytical journeys that can persuade, inform, and inspire. As American writer, John Ernst Steinbeck Jr. stated, "People don't take trips; trips take people."

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
