# OpenReview forum: "Changing Lanes: Towards Symbolizing Visualizations"
_IEEE.org/2022/Workshop/altVIS — Reject_

### Official Review · Reviewer_sBQe · 2022-08-05

**Review:**


The final sentence of the abstract poses an interesting question ("By focusing more on intention rather than just precision, can we symbolize visualizations and change the lanes for both the researcher and practitioner community?"). Unfortunately, the rest of the article doesn't really address this in a satisfactory way.

The importance of factors beyond precision has been discussed in prior work such as [Why Shouldn’t All Charts Be Scatter Plots? Beyond Precision-Driven Visualizations](https://ieeexplore.ieee.org/document/9331277).

# Originality

The authors of this have recently written a book (*Functional Aesthetics for Data Visualization*): paper copies of this will be officially published in a few days time, but electronic versions are already available. The paper cites the book once:

> The use of icons in charts depends on various factors, including task, how representative they are of the underlying data, and their general recognizability [7].

**However, given the extent of similarities between the book and the paper, I consider this to be insufficient, and in my opinion this alone is enough to require rejection of this paper.**


Most of the figures seem to have been reproduced from the book:

* Figure 1 of this paper is extremely similar to Figure 4.1 of the book; there are some minor modifications (the Korean flag has been deleted, the "flatten the curve" graphic added, and the labels removed)

* Figure 2 of the paper is essentially the same as Figure 4.3 of the book

* Figure 3 of the paper is essentially the same as Figure 1.20 of the book

* Figure 5 of the paper is essentially the same as Figure 12.6 of the book

Figure 4 and 7 seem to be new.


There are also substantially similarities in the text.

Arbitrarily selecting section 4 for comparison:


> A map is another unique form of symbolic representation but is spe-
cialized for space. It depicts spatial relationships with objects in the
geographic and physical landscape. By scaling down to a natural size
conducive for reading, a map allows the viewer to see far more infor-
mation in one glance than could ever be possible from ground level
without the need for any pictorial realism.

This is from p.16-17 of the book.

> Maps have inspired the early roots of information visualization by
depicting information as a communication process [6]. While details
of these depictions vary, all maps share a basic structure with an in-
formation source identified by the cartographer who determines what
and how to represent that information to the reader.

This is from p.18 of the book.

> The map serves
as one of many potential representations of information in space that
allows the reader to access its meaning. The process of communi-
cating spatial information has evolved as a system with information
inputs, transmission, and reception of that information to the intended
audience.

I think this is a new addition, but it is essentially without substance.


> Cartographers have found ways to make maps more legible and
useful. They often simplify or eliminate less semantically important
features, exaggerate more important ones, and resolve visual clutter to
improve information quality. This is a process known as generalization
(as shown in Figure 3).

This is from p. 18 of the book.

> Route maps have succinctly applied various
forms of generalization to effectively provide navigational direction.
Generalization helps emphasize what is important and deemphasize the
unimportant, showing points of interest, turns, and interactions such
as pan and zoom.

This is from p. 20 of the book.


> Mapmakers make explicit decisions about which
aspects of the route are most relevant for a navigator to understand
and follow the route. They use a variety of cartographic generalization
techniques to help improve the clarity of the map and emphasize only
the most important information.

This is from p. 22 of the book.



# Understandability

I find it hard to follow the argument of this paper.

It doesn't really feel cohesive (as I realized after writing my initial review, this is probably because it consists of sections of text that have been copied and pasted from a book).

Instead, it seems like a grab-bag of familiar observations: different registers are appropriate for different communication contexts, and pie charts are not always bad (Section 2); simplified or schematic diagrams are useful in communicating concepts (Section 3); cartographic generalization introduces minor inaccuracies in order more clearly communicate some important aspects (Section 4); transport signs need to be standardized and quickly comprehensible (Section 5); unfamiliar visualizations or interactions require explanations (Section 5).

Figure 5 claims to change the register of a chart by transitioning from an area chart to a line chart, via a chart with both a line and a light fill. I don't really see how this is a difference in register or degree of formality. It seems a clearer example would be to add or change annotations and explanatory text. I also do not understand what is meant by claims like "the line chart is less comfortable to the audience".

The description of Figure 2 states: "As an icon, it has been abstracted into two rounded area charts". However, the icon looks like a line chart rather than an area chart - the area under the curve is not filled. Also, I think that the horizontal "healthcare capacity" line is a key part of the iconic "flattening the curve" graphic, but this is missing from Figure 2b. And whereas the process of converting to an icon would be expected to simplify and remove clutter like grid-lines, a dashed background grid has been *added* to Figure 2b.


## Figure 7

I don't really see how Figure 7 relates to the previous examples (except that it includes a donut chart, which Section 2 mentions).

I also find this dashboard hard to read. There is an apparently meaningless horizontal blue bar at the top-right; the left edge of this is aligned with the left edge of the text, which initially gives the mistaken impression that the dashboard is divided into separate left and right parts. It also interrupts the flow of the heading "Primary Cafe is Overutilized", by dividing this text into two parts.  If the purpose of the bar is to create a visual association between the dark blue color of the primary cage and the word "overutilized", then this could have been better achieved by writing just this word in dark blue.

The list of takeaways says "note how “MPH” can be inferred", but that is only true if you know what country the sign is from and what units they use for their speed signs: either 50 mph (80 kph) or 50 kph (31 mph); could be a reasonable limit depending on the road type. Similarly, its impossible to tell what the 346 and 87 written on the bars means without prior knowledge (number or value of sales in some time period? average wait time?).

I have no idea what the radial plots centered on each cafe show.

# Other

All quotations should have references; currently some do not (e.g., "The ordering of the tasks does not result in a precise prescription for displaying data, but rather is a framework within which to work")


**Conflicts:**

None

**Review Inclusion:**

No

**Sufficiently Alt:**

Yes

**Superlative:**

Most self-plagiarised

---

### Official Review · Reviewer_fUbv · 2022-08-08

**Review:**

In this work, the authors present a position suggesting that visualization may benefit from greater use of symbolization. This position is supported by a series of examples (flatten the curve, maps, and road signs). While the ultimate call to action derived from this stance is somewhat pedestrian---and well supported by other research (for instance systematizing charts (in a dashboard) is covered in Qu & Hullman's "Keeping multiple views consistent")---it is still an interesting perspective overall. I generally like this presentation of the material, however, I wish that the argument supporting it was somewhat tighter/more focused, as the current presentation feels a bit disjoint, which impedes the clarity of the intended provocation.

A few notes

- I'm not sure I entirely buy the assertion that semantics is not already a first-class citizen in a lot of visualization design. For instance, Nigel Holmes' monstrous costs does specifically utilize a set of signs readers are likely familiar with to both facilitate a particular reading of that data, but also just to make that material more engaging. How does the data visualization task inform the role that symbolization should take (eg should it just occur in presentational visualization or could EDA find a use for it as well)? How might visualization systems or tools (like Tableau) more explicitly surface semantics?
- I think an important differentiator between some of the discussed symbol sets is the role that they play and the influence of their context on the interpretation of that sign. In my own experience with it, the flatten the curve icon was used more as a rhetorical device (traversing into sloganeering territory) than for decision making, whereas maps utilize a variety of different visual semantics to facilitate specific tasks. Finally, roadsigns create an extreme incentive for their users to understand them, both in terms of safety but also in terms of the law. The symbols we have chosen to represent many traffic interactions are deeply arbitrary (cf the 99 percent invisible history of road signs), and it is only through repeated and legally enforced convention that they have become culturally ingrained. How might we understand which symbolization is appropriate for which contexts? How might we understand when external pressure appropriately guides the selection of a particular sign as an appropriate signifier?
- Semiotics has a long history with visualization (Ex Bertin's book), and in arguing for greater use of signs, I would have liked to see a closer relationship with some explorations found in prior literature. For instance, "Understanding visualization: A formal approach using category theory and semiotics" / "Visual semiotics & uncertainty visualization: An empirical study". Similarly, it might be good to locate this work with respect to prior studies on the use of icons _in_ visualization, such as isotypes. How does this work relate to these prior works?
- I like seeing this work as a register exercise, however, I wish that the paper spent more time describing what that means. If I'm not mistaken the structure of the paper seems to intend to follow the structure suggested later (with simple elements coming first etc, or the tone shift between "not my dessert" and "flattening the curve")? If so it might be good to highlight this self-similarity explicitly. In either case, it would be good to define the term more clearly.
- There are several assertions that would probably be well backed up by citation. For instance "Lines may be hard for some readers to navigate", "A common issue with BANs is that they are not formatted well enough to be effective", etc.
- Typo? "much aligned and" -> "much maligned and"

**Conflicts:**

I do not believe I have any conflicts.

**Review Inclusion:**

No

**Sufficiently Alt:**

Yes

**Superlative:**

Most Iconic

---

### Official Review · Reviewer_41nW · 2022-08-23

**Review:**

Paper summary: The paper suggests symbolising visualisations using icons and universal agreed-upon symbols to make the visualisations more accessible to the public. I think the paper’s idea highlights a hot relevant topic that is very current and in need. Focusing on the general public and considering varying graphical and numeracy literacy levels is a persisting challenge. I think the approaches suggested to tackle these challenges using means people are familiar with (e.g., road signs) are creative. Overall, I would argue that the paper is interesting and considers the design from a viewpoint often neglected, but I doubt it fits the alt.VIS criteria. I feel the paper can easily fit in any other venue. Nevertheless, I suggest the authors clarify the paper’s main goal and link it to previous work.


Pros:

1.	The paper focuses on a group of the population that has received less attention than experts in visualisations.
2.	The figures and illustrations were helpful.

Cons:

1.	The introduction and paper motivation need some clarification. For example, the following statement is unclear: “By considering semantics as part of the visualisation authoring process, we can take a visual abstraction of data, agree on its meaning, and put it to use as a shared symbol”. Does this mean using icons and symbols to illustrate the central message of the visualisation instead of the graph? Does the icon replace the graph? If so, I think this is interesting. However, I could not see that later in the paper. Perhaps the primary goal can be clarified more in the introduction.

2.	I could not understand some of the idioms used in the paper. For example, “we perhaps should not be so quick to dish on our dessert.”, “Not my dessert”, “I love you” in ASL”. Perhaps for inclusivity, give more context around them.

3.	The paper states that the primary goal of symbolising visualisation is to create accessible visualisations, which is an important goal. However, the connection between each recommendation and the main goal is not apparent. To fix: expand on how each approach will help achieve a more accessible design.

4.	The idea of using icons and symbols is great, but what about cultural differences and familiarity? I feel that needs addressing as a limitation of the approach.

5.	Another limitation that needs to be addressed is objectivity. In section 3, the suggestion to use the ambiguity of the data collected to highlight the limitations of the health care system is unclear to me. Are you suggesting that to be the main takeaway and symbolise that idea? If so, wouldn’t that be too leading, make the chart less objective, and even seem a bit patronising?

6.	I think the viewpoint of using the four suggested approaches to create accessible visualisations is valuable. However, references to existing work are missing. For instance, using semantically resonant colours: https://dl.acm.org/doi/10.5555/2600534.2600590, decluttering and focusing and highlighting technique: https://visualthinking.psych.northwestern.edu/projects/DeclutterFocus/Ajani_Declutter_2021.pdf.


7.	For the paper motivation, I suggest adding the following literature: International Picture Language By Otto Neurath https://monoskop.org/images/e/ec/Neurath_Otto_International_Picture_Language.pdf, whose work is about creating a universal language using icons, which is highly connected to the paper’s goal, Beyond Memorability: Visualisation Recognition and Recall Supplemental Material, https://vcg.seas.harvard.edu/files/pfister/files/infovis_submission251-camera.pdf , and finally, Casual information visualisation: Depictions of data in everyday life, https://ieeexplore.ieee.org/document/4376134




**Conflicts:**

I have no conflicts of interest with the authors

**Review Inclusion:**

No

**Sufficiently Alt:**

No

**Superlative:**

ambitiously creative

---

### Official Review · Reviewer_LBRx · 2022-08-24

**Review:**

The research directs us to an interesting visualization of meaningful icons, symbols and imagery. However, the quality of the visualizations is low and not ready for publications.

**Conflicts:**

NA

**Review Inclusion:**

No

**Sufficiently Alt:**

Yes

**Superlative:**

Lowest resolutions

---

### Official Review · Reviewer_Efqf · 2022-08-31

**Review:**

Alt-Meta Review:

I regret to inform the authors that this paper has been rejected from alt.vis 2022. While the topic presented is interesting, there are many things that are missing from the paper. Additionally, it is not alt-y enough, but I encourage the authors to consider submitting to another venue. If the authors plan to revise and resubmit the paper, I strongly recommend incorporating the suggestions of the other reviewers which aptly highlight necessary additions like articulating a clearer goal for the paper, citing work about the relevance of context in interpretation, and including a related work discussion regarding the highly relevant field of semiotics.

**Conflicts:**

None that I am aware of.

**Review Inclusion:**

No

**Sufficiently Alt:**

No

---

### Decision · Program_Chairs · 2022-08-31

Reject